# Optimizing Absorption for Intranasal Delivery of Drugs Targeting the Central Nervous System Using Alkylsaccharide Permeation Enhancers

**DOI:** 10.3390/pharmaceutics15082119

**Published:** 2023-08-10

**Authors:** Stuart Madden, Enrique Carrazana, Adrian L. Rabinowicz

**Affiliations:** 1Neurelis, Inc., San Diego, CA 92121, USA; ecarrazana@neurelis.com (E.C.); arabinowicz@neurelis.com (A.L.R.); 2John A. Burns School of Medicine, University of Hawaii, Honolulu, HI 96813, USA

**Keywords:** absorption, central nervous system, dodecyl maltoside, intranasal, Intravail, nasal cavity, rescue therapy, route of administration, solubility, tetradecyl maltoside

## Abstract

Intranasal delivery of drugs offers several potential benefits related to ease of delivery, rapid onset, and patient experience, which may be of particular relevance to patients with central nervous system (CNS) conditions who experience acute events. Intranasal formulations must be adapted to address anatomical and physiological characteristics of the nasal cavity, including restricted dose volume, limited surface area, and barriers to mucosal absorption, in addition to constraints on the absorption window due to mucociliary clearance. Development of an effective formulation may utilize strategies including the addition of excipients to address the physicochemical properties of the drug within the constraints of nasal delivery. Dodecyl maltoside (DDM) and tetradecyl maltoside are alkylsaccharide permeation enhancers with well-established safety profiles, and studies have demonstrated transiently improved absorption and favorable bioavailability of several compounds in preclinical and clinical trials. Dodecyl maltoside is a component of three US Food and Drug Administration (FDA)–approved intranasal medications: diazepam for the treatment of seizure cluster in epilepsy, nalmefene for the treatment of acute opioid overdose, and sumatriptan for the treatment of migraine. Another drug product with DDM as an excipient is currently under FDA review, and numerous investigational drugs are in early-stage development. Here, we review factors related to the delivery of intranasal drugs and the role of alkylsaccharide permeation enhancers in the context of approved and future intranasal formulations of drugs for CNS conditions.

## 1. Introduction

Intranasal drug delivery is a promising option for an increasing number of drugs for systemic absorption and, importantly, for targeting the central nervous system (CNS). Formulating treatments for intranasal delivery must take into account the unique features of nasal anatomy and physiology, which offer potential benefits and present specific challenges to be addressed by a successful formulation. The intranasal route is noninvasive, and prefilled devices can offer the benefits of quick administration for acute events and ease of use by nonmedical caregivers in the community, including the potential for self-administration [1]. Surveys report positive impressions and satisfaction with existing intranasal formulations among patients and caregivers and preference over other routes of delivery, such as rectal administration [2]. Intranasal formulations also offer potential benefits in terms of favorable pharmacokinetics, including potentially high bioavailability while bypassing variable gastrointestinal absorption and first-pass metabolism [3]. At the same time, intranasal delivery necessitates a specific formulation that is adapted to the unique characteristics of the nasal anatomy and physiology, including a surface area that limits dosing volume, permeability of nasal mucosa, mucociliary clearance, and issues of tolerability from either the drug substance or the formulation itself [3].

A growing number of intranasal drug formulations for the treatment of neurologic conditions have been developed in recent years. One treatment, esketamine nasal spray, is administered regularly under the supervision of a healthcare provider as an ongoing adjunct treatment for depression [4]. However, many are treatments for acute conditions, such as triptans dihydroergotamine and zavegepant for migraine, naloxone and nalmefene for opioid overdose, and diazepam and midazolam for seizure cluster in epilepsy [5,6,7,8,9,10,11,12,13]. In the first half of 2023 alone, the intranasal formulations of zavegepant (for migraine) and nalmefene (for opioid overdose) were approved by the US Food and Drug Administration (FDA), and naloxone nasal spray has been approved for nonprescription use in the US and is being made available directly to the public [10,13,14]. While intranasal formulations are becoming increasingly available, there remains a lack of awareness among researchers, clinicians, and patients of their unique characteristics for intranasal administration and drug delivery. Understanding intranasal delivery and strategies for the development of intranasal drug formulations will help clinicians assess and differentiate intranasal formulations and ultimately facilitate working with their patients to choose the most appropriate treatment option. In this review, we address aspects of nasal anatomy and physiology that are particularly relevant to intranasal delivery of CNS drugs. Within the broad context of strategies to facilitate intranasal absorption, which also include nanocarriers [15], gels [16], and devices [17], we focus on the development of alkylsaccharides as permeation-enhancing excipients and the data supporting their clinical use in a growing number of intranasal formulations that are increasingly available in different regions.

## 2. Challenges of Nasal Drug Delivery

### 2.1. Nasal Anatomy and Physiology

Structures of the nose heat and humidify air for respiration; clear foreign particles, including pathogens; and facilitate olfaction [3]. The two nasal cavities are irregularly shaped internal structures lying between the roof of the mouth and the skull bones encasing the bottom of the brain (Figure 1) [18]. They are separated by the septum, while the outer cavity walls are folded to form three turbinates on each side [18]. The nasal cavity contains four general areas: the vestibule, the respiratory mucosa, the olfactory mucosa, and the nasopharynx-associated lymphatic tissue [3].

The central portion of the nasal cavity is covered by the respiratory and olfactory mucosa and represents the area most relevant for drug absorption after intranasal administration. The respiratory mucosa overlays the inferior and middle turbinates, as well as the lower part of the septum [3]. It is the largest portion of the nasal cavity and is highly vascularized by a rich bed of blood vessels containing pores (fenestrations) in the vessel walls [3,18]. The characteristics of these vessels make them a prime potential site for drug absorption into general circulation. In addition, the presence of the turbinates increases the overall surface area of the nasal cavity to approximately 160 cm^2^, which is available for drug absorption.

The olfactory mucosa constitutes ≤10% of the nasal cavity surface area and covers the upper portion of the cavity [3,18]. This region contains olfactory sensory neurons, which are the only neurons of the CNS on the surface of the body exposed to the external environment [18]. This exposure offers the potential for direct delivery of CNS drugs to the brain, although the location may present an obstacle to the efficient deposition of intranasal drugs. The trigeminal nerve, lying beneath the nasal mucosa, also innervates the nasal cavity and offers the potential for direct-to-brain delivery of drugs [3,18]. In this context, it is important to note the differences in nasal anatomy and physiology between humans and animal (e.g., rat) models used to study nasal delivery systems [19]. These differences, such as a proportionally larger olfactory region in rats, may limit the direct applicability of findings from animal models to humans but can be used to assess the rank order of nose-to-brain delivery of analogs, for example [20].

### 2.2. Considerations for Nasal Drug Formulations

#### 2.2.1. Anatomic and Physiologic Constraints

Although the nasal cavity may effectively capture intranasally delivered medications, it also creates a finite volume for dose delivery. The optimal volume for a dose of intranasal drug is no more than 100 to 150 µL per nostril [1,18]. Larger volumes risk the partial loss of drug dose either anteriorly through the nostrils or posteriorly into the nasopharynx, which is then swallowed and absorbed as an oral drug, subject to variability in bioavailability based on fed state and first-pass metabolism [3]

Second, mucus coating the epithelium of the nasal cavity acts as a defense against pathogens and particulates. Beating cilia continuously move mucus into the nasopharynx, and it is estimated that the mucus layer in the nasal cavity renews every 10 to 20 min [21]. This may reduce residence time in the nasal cavity and affect drug absorption.

Third, the mucosal lining of the nose is sensitive to irritation by drugs or other compounds, which can lead to inflammation, secretions, itching, sneezing, and pain [17]. This may constrain the choice of drug for intranasal delivery or the addition of certain excipients [22]. Although there is concern that seasonal allergies or rhinitis may negatively affect intranasal drug delivery and absorption, several studies of different intranasal formulations have demonstrated a similar extent of absorption or bioavailability between patients with and without allergic rhinitis [23,24,25]. No clinical impact of rhinitis or seasonal allergies was observed in separate safety studies of zolmitriptan and diazepam nasal sprays [26,27]. Finally, for intranasal deposition, droplets of 10 to 120 µm in diameter may be ideal because smaller droplets may be inhaled and deposited in pulmonary tissue, while larger droplets may deposit predominantly in the anterior nasal cavity [28,29].

#### 2.2.2. Formulation and Physiochemical Constraints

An optimal intranasal dose of no more than 100 to 150 µL translates into a requirement for adequate drug solubility in a fixed and relatively small dose volume. Physiochemical properties of a drug, such as its lipophilic-hydrophilic properties, and its molecular weight and size affect how it can be formulated for intranasal delivery and its absorption across nasal mucosa to ensure adequate and consistent bioavailability [30]. In general, small, lipophilic drugs are more easily absorbed across the nasal mucosa; however, they are less soluble in the aqueous delivery systems commonly used in nasal formulations [1,29,31].

Excipients may be used to modify drug solubility, membrane permeation capacity, mucosal tolerability, mucosal adhesion, or formulation stability [30]. These agents include buffers, cosolvents, absorption enhancers, mucoadhesives, and viscosity enhancers, which must balance the desired effect with the potential for toxicity and/or irritation in the formulation. For example, decreasing pH may increase the intranasal absorption of some drugs and decrease it for others, while sensitizing the nasal cavity as the formulation deviates from the natural pH of the nasal cavity (pH 5.5–6.5) [30,32,33], which carries a risk of local irritation [1].

#### 2.2.3. Delivery Device Constraints

The intranasal delivery system must create reproducible parameters of dose delivered, droplet size distribution, and spray pattern and plume geometry to ensure consistent dosing to the intended target tissue [34]. Several delivery systems are available, from droppers and mucosal atomization devices (MADs) attached to syringes to prefilled unit-dose or unidose pumps, multidose pumps, and propellent-activated devices. With droppers and MADs, administration may be inconvenient, requiring awkward positioning of the patient, and precise dosing may be difficult [31]. Unidose pumps are single-use devices delivering a precise dose of the drug with simple actuation, whereas multidose pumps have the capability for repeat use during the course of treatment. A propellent-powered device is also being studied with a limited number of CNS drugs [35,36].

The delivery system must provide ease of use for patients who self-administer treatment or for care partners responsible for administration. The device should function reliably in any orientation; have proven biocompatibility so as not to cause irritation to the lining of the nostrils; and, for delivery to incapacitated patients, be able to be used without their active participation [1,31]. Unidose delivery systems are optimized for easy handling and intuitive use. Finally, the drug–device combination product should have an acceptable drug shelf life at controlled room temperature and be sufficiently robust to withstand shipping.

## 3. Focus on Absorption Optimization

To overcome the constraints imposed by nasal anatomy and physiology, including small volume, continual mucociliary clearance, and potential low drug permeability through the nasal mucosal membranes, intranasal formulations may use two broad strategies to optimize drug absorption and bioavailability: increase residence time in the nasal cavity or increase the rate of absorption across the mucosal membrane [30]. Both strategies employ specialized excipients and may be used together in a single formulation. Increasing drug permeation carries the advantages of increased bioavailability with lower drug load within the dose volume, which in turn may reduce the solubility burden on the formulation. Increased permeation may also be associated with less variability in the total dose delivered and has the potential to deliver higher-molecular-weight molecules that are typically poorly absorbed intranasally. An ideal permeation enhancer is compatible with the drug, soluble and stable in a range of solvent systems, and commercially available with appropriate labeling by a regulatory body (Figure 2).

### 3.1. Biochemistry of the Alkylsaccharides Dodecyl Maltoside and Tetradecyl Maltoside 

Alkylsaccharides are a family of sugars characterized by covalent linkages to ≥1 alkyl chain (Figure 3) [37], several of which have been developed as transmucosal delivery enhancement agents. They are classed as nonionic surfactants and used in food and personal care products. In particular, dodecyl maltoside (DDM; 511 Da) and tetradecyl maltoside (TDM; 539 Da) are examples of Intravail^®^ (Neurelis, Inc., San Diego, CA, USA) alkylsaccharide absorption enhancers [38]. These excipients are considered safe, nontoxic, nonmutagenic, tasteless, odorless, and nonsensitizing (up to 25% concentration in the Draize test); they are designated as Generally Recognized as Safe (GRAS) for oral consumption and meet all the requirements listed in Figure 2 [37]. Sucrose esters have a no-observed-effect level (NOEL) of 2000 mg/kg body weight and a World Health Organization oral allowable daily intake (ADI) of ≤40 mg/kg body weight, supporting their use as an excipient [39]. In the body, alkylsaccharides are metabolized to their constituent sugars and fatty acids, and ultimately to carbon dioxide and water [40]. DDM is soluble in oils and water, making it appropriate for formulations of hydrophilic and hydrophobic compounds [40]. DDM is compatible with routine liquid formulation and dispensing processes for ease of scale-up and production. Intravail is composed of synthetic pure chemical entities prepared under good manufacturing practices (GMP) in Europe and Asia to ensure uniform composition between lots and below-threshold levels of residual reactants or solvents [37].

### 3.2. DDM and TDM Mechanism of Action

Two possible mechanisms of action have been proposed for Intravail excipients’ effect of increased transmucosal transport. The first involves paracellular transport via transient loosening of tight junctions between epithelial cells by DDM to permit mucosal barrier permeation, which has been demonstrated in several human cell lines and epithelial tissue [38]. A study in the rat demonstrated a transient permeation effect on nasal epithelia after the application of TDM with successful transport of molecules up to 22 kDa (human growth hormone analog somatropin) [41]. The temporary opening and then closing of tight junctions was observed through the time-dependent exclusion of successively smaller molecules after treatment (Figure 4). When calcitonin (4 kDa) was administered 60 and 120 min after TDM, absorption was reduced but not halted, indicating that the tight junctions remained sufficiently open to allow passage of the small molecule. However, when somatropin (22 kDa) was administered 60 and 120 min after TDM, absorption was prevented, suggesting that the tight junctions had closed sufficiently to exclude the larger compound. The second mechanism of action involves transcellular transport across the mucosa via vesicle-mediated mechanisms, as observed in electron micrographs of the rat nasal septum tissue treated with TDM (Figure 5) [42]. DDM has also demonstrated vesicular transport effects in a tissue explant system [38].

### 3.3. Preclinical Studies of DDM and TDM

In animal studies of intranasal absorption, the use of TDM was associated with increased absorption of calcitonin (3.5 kDa), enoxaparin (4.5 kDa), and dalteparin (5 kDa) [41,43,44]. The calcitonin study measured plasma levels in rats over time after nasal administration of formulations containing saline, a medium-chain-length alkylsaccharide, octylmaltoside, or TDM [43]. The lipophilic characteristics of an alkylsaccharide are related to alkyl chain length, and lipophilicity increased with increasing chain length. Saline and octylmaltoside (C_8_ chain length) formulations resulted in little to no absorption of calcitonin, whereas TDM (C_14_ chain length) formulations (0.125–0.25%) resulted in increased calcitonin plasma levels, peaking 7.5 to 10 min after administration [43]. Using low-molecular-weight heparin composed of enoxaparin and dalteparin, researchers tested the absorption-enhancing effect of TDM (0.25%) in rats. Intranasal administration of a formulation containing TDM resulted in a significant increase in peak plasma concentration and area under the curve for anti–factor Xa activity (a surrogate for heparin absorption) and higher bioavailability compared with saline alone, as well as reversibility [44]. Building on those findings, researchers then tested formulations of enoxaparin and four different alkylsaccharides, including DDM and TDM [45]. Intranasal administration with these alkylsaccharides resulted in dose-dependent and chain length–dependent increases in the absorption of enoxaparin, while all had a reversible effect [45]. Subsequent animal studies have assessed the bioavailability of several molecules (≤30 kDa) delivered intranasally with different concentrations of TDM [37]. The studies demonstrated two notable qualities: (1) a dose-dependent relationship between TDM concentration and bioavailability of the drug and (2) an inverse relationship between molecular weight and drug bioavailability (Figure 6) [37]. For some molecules, bioavailability after intranasal delivery with DDM or TDM is comparable to intravenous delivery [37,46].

### 3.4. DDM: Currently Approved Treatments

To date, three intranasal medications containing DDM have been approved by the US Food and Drug Administration (FDA). They are treatments for migraine, seizure cluster in epilepsy, and opioid overdose. Clinical trials of these treatments support the favorable safety profile of DDM as an excipient. Currently, no intranasal treatments containing DDM are approved by the European Medicines Agency for use in Europe.

#### 3.4.1. Sumatriptan Nasal Spray with Intravail (Tosymra^®^)

Sumatriptan nasal spray with DDM (Tosymra^®^, Upsher-Smith Laboratories, Maple Grove, MN, USA) is indicated for the treatment of acute migraine headaches with or without aura in adults [9]. A randomized crossover study compared intranasal sumatriptan 10 mg containing 0.20% DDM with the earlier commercially available intranasal sumatriptan 20 mg without DDM (n = 18 healthy volunteers) [47]. The earlier formulation is characterized by low bioavailability (17%) [8] and slow absorption, whereas the newer formulation with DDM has a bioavailability of 58% to 87% [9]. In the crossover study, the median time to maximum plasma concentration (C_max_) for the DDM formulation was 10.2 min, while for sumatriptan 20 mg without DDM, the median time was 2 h. The concentration-time profile of the formulation without DDM registered two peaks. The researchers interpreted the early peak in plasma levels as an indication of nasally absorbed drug, while the second later peak was likely due to most of the dose being swallowed and absorbed through the gastrointestinal tract (Figure 7). In contrast, the DDM formulations demonstrated a single early peak of absorption, suggesting nasal absorption [47].

The efficacy, safety, and tolerability of sumatriptan plus DDM were assessed in phase 2, a double-blind, 2-period, placebo-controlled study of sumatriptan 10 mg with 0.20% DDM [48]. The study, which enrolled 107 patients with migraine, met its primary endpoint: the proportion of patients free from headache pain at 2 h postdose was statistically significantly higher in the sumatriptan with DDM group than in the placebo group (last observation carried forward: 43.8% vs. 22.5%, *p* = 0.044; observed cases: 43.8% vs. 20.5%, *p* = 0.025). Treatment-emergent adverse events (TEAEs) were reported in 9/93 (9.7%) patients, and 7/93 (7.5%) reported a TEAE that was considered to be treatment-related. The most commonly reported TEAEs were dysgeusia (3 patients) and application site pain (2 patients) [48].

In the same report, researchers described a 3-way crossover study comparing pharmacokinetic parameters of intranasal sumatriptan plus DDM and 4 or 6 mg sumatriptan administered subcutaneously in 78 healthy volunteers. Intranasal sumatriptan with DDM provided significantly faster time to maximum concentration (t_max_) than subcutaneous dosing (10 vs. 15 min; *p* < 0.0001), with similar overall exposure, measured as the mean area under the curve (AUC), between intranasal sumatriptan 10 mg with DDM and the 4- mg subcutaneous dose. TEAEs occurring in ≥10% of patients in the trial overall study were dysgeusia (19%, n = 15 [13 occurred with the intranasal formulation]), headache (18%, n = 14 [5 intranasal]), nausea (15%, n = 12 [2 intranasal]), paresthesia (15%, n = 12 [0 intranasal]), and dizziness (12%, n = 9 [2 intranasal]). In an open-label, long-term, repeat-dose safety study of the intranasal formulation of sumatriptan 10 mg with DDM, 51 of 167 (30.5%) patients reported application site pain (including nasal burning or stinging), usually of mild severity [49]. Incidences of application site reaction and irritation were 5.4% and 4.2%, respectively. It is important to note that these studies did not assess the safety of DDM alone compared with sumatriptan plus DDM, so the contribution of active drug vs. excipient to treatment safety and tolerability cannot be determined.

#### 3.4.2. Diazepam Nasal Spray (Valtoco^®^)

Dodecyl maltoside is also a component of diazepam nasal spray (Valtoco^®^, Neurelis, Inc., San Diego, CA, USA), which is indicated for the acute treatment of intermittent, stereotypic episodes of frequent seizure activity (i.e., seizure clusters, acute repetitive seizures) that are distinct from a patient’s usual seizure pattern in patients with epilepsy 6 years of age and older [5]. It includes vitamin E to increase diazepam solubility without the use of potentially irritating organic cosolvents that would be used in conventional aqueous systems [50]. An early crossover pharmacokinetic study compared two intranasal diazepam 10-mg formulations that contained DDM with commercially available intravenous diazepam in healthy volunteers (n = 24). Diazepam nasal spray solution had an absolute bioavailability of 97% relative to the intravenous formulation with similar variability in exposure, as measured by AUC [46]. 

A phase 1 open-label crossover study in healthy volunteers (n = 48) assessed the bioavailability of diazepam nasal spray compared with oral and rectal diazepam [51]. Intranasal delivery was associated with less interpatient variability in bioavailability (as measured by AUC) than rectal delivery, while overall bioavailability was slightly less. The pharmacokinetics of diazepam nasal spray was also studied in patients with epilepsy with seizure clusters [52]. There was little impact of the epileptic state (ictal/peri-ictal vs. interictal) on pharmacokinetics or safety [52]. The study also assessed levels of DDM in a subset of patients (n = 25) with epilepsy after treatment with diazepam nasal spray [53]. At any time point, ≤5 patients had detectable DDM plasma concentrations (>500 pg/mL; maximum concentration 741 pg/mL), with levels returning to undetectable by 1.75 h postdose, confirming that residual plasma concentrations of DDM are low and transient following use of diazepam nasal spray [53].

The safety of diazepam nasal spray was assessed in a long-term, open-label, repeat-dose phase 3 study of patients aged 6 to 65 years with epilepsy and seizure clusters (n = 163 safety population) [50]. Overall, the safety profile of diazepam nasal spray was comparable to that of rectal diazepam. Over the course of the 12-month study, 134 (82.2%) patients reported any TEAE. Treatment-related TEAEs were reported by 30 (18.4%) patients: events reported by ≥3 patients were nasal discomfort (n = 10 patients [6.1%]); headache (n = 4 [2.5%]); and dysgeusia, epistaxis, and somnolence (n = 3 each [1.8%]). There were no clinically significant toxicities of the nasal cavity. Nasal irritation was assessed by a trained observer. Most patients experienced no sign of irritation (764 of 781 tests), with the remaining cases of irritation being transient. No significant olfactory changes were observed by the NIH Toolbox Odor Identification Test. Finally, rates of TEAEs, serious TEAEs, and treatment-related TEAEs were similar between patients with and without seasonal allergies/rhinitis [50]. As with the studies of sumatriptan with DDM, this study did not assess the safety of DDM alone; thus, the relationship to DDM cannot be determined.

Finally, surveys of patients with seizure clusters and their caregivers who used intranasal diazepam rescue therapy delivered in unidose pumps reported favorable impressions of the treatment. Patients and caregivers found it comfortable to carry and use outside the home, and they preferred the convenience of the nasal spray over rectal diazepam [2]. Of note, a study of a prefilled intranasal spray pump (naloxone) found it to be easier to use than an assembled MAD or an intramuscular autoinjector [54]. The intranasal device was also associated with a shorter time to successful delivery (16 s) than either the MAD (113 s; *p* = 0.012) or the autoinjector (58 s; *p* < 0.001) [54].

#### 3.4.3. Nalmefene Nasal Spray (Opvee^®^)

In May 2023, nalmefene nasal spray with DDM (Opvee^®^, Indivior Inc., Chesterfield, VA, USA) received FDA approval for healthcare and community use as an emergency treatment for known or suspected opioid overdose in adults and pediatric patients 12 years of age and older. In a preclinical study assessing three absorption enhancers including DDM in an animal model, absolute nalmefene bioavailability with 0.5% and 0.25% DDM was 76.5% and 71.0%, respectively, a significant increase (*p* < 0.05) over the bioavailability of nalmefene administered alone (47.7%) [55]. Nalmefene plus DDM reported good safety in nasal ciliotoxicity models [55]. A phase 1 pharmacokinetic study of nalmefene in healthy volunteers (n = 14) compared four formulations: intranasal nalmefene 1.5 mg, intranasal nalmefene 3 mg, intranasal nalmefene 3 mg plus 0.25% DDM, and intramuscular nalmefene 1.5 mg [56]. Median t_max_ of intranasal nalmefene 3 mg was 2 h. The addition of DDM reduced t_max_ to 15 min and more than doubled C_max_ (4.45 vs. 1.99 ng/mL for nalmefene 3 mg). DDM had no apparent effect on nalmefene half-life. The onset of action with nalmefene with DDM was comparable to the intramuscular injection, with a higher C_max_. TEAEs were mild, with no apparent changes in olfaction [56].

Three clinical trials (2 pharmacokinetic trials and 1 pharmacodynamic study) support the safety of nalmefene nasal spray. The most common treatment-related TEAEs (>5%) associated with nalmefene 2.7 mg nasal spray in healthy volunteers (n = 150) were nasal discomfort (28.7%), headache (26.7%), nausea (16.7%), dizziness (9.3%), hot flush (8.0%), and vomiting (6.0%) [13]. These studies did not assess the safety of DDM alone; thus, the relationship to DDM cannot be determined.

### 3.5. DDM: Future Directions

#### 3.5.1. Epinephrine Nasal Spray (Neffy™)

Intramuscular injection of epinephrine is the standard of care for the treatment of severe allergic reactions, including anaphylaxis. Epinephrine nasal spray (neffy™, ARS Pharmaceuticals, Inc., San Diego, CA, USA) is under development as a noninvasive alternative to injection for use in community settings [57]. Epinephrine nasal spray contains 0.275% DDM, which was chosen based on optimal epinephrine bioavailability in phase 1 studies [58]. A recent publication presented an integrated analysis of data from four phase 1 studies comparing the pharmacodynamic and pharmacokinetic properties of intramuscular epinephrine 0.3 mg injection either manually or with autoinjectors and epinephrine 1 mg nasal spray. Within the context of the emerging understanding that epinephrine pharmacokinetics can vary among different injection devices, epinephrine nasal spray with DDM demonstrated a mean C_max_ that was similar to manually injected epinephrine, with a shorter median t_max_. Overall, the nasal spray produced similar or more pronounced pharmacodynamic changes compared with the injected formulations [57]. Epinephrine nasal spray is currently under review with the FDA.

#### 3.5.2. Other Treatments

Naltrexone with DDM (Indivior, Chesterfield, VA, USA) is in development as a nasal spray for the treatment of patients with alcohol use disorder. Based on a pharmacokinetic study in healthy volunteers that showed a decrease in t_max_ from 30 to 10 min and a ~3-fold increase in C_max_ with the addition of DDM to intranasal naltrexone [59], a phase 2 study is underway. The randomized, double-blind, placebo-controlled trial will evaluate the effects of naltrexone nasal spray on alcohol use disorder, with preliminary results expected in 2023 [60,61]. Finally, other intranasal formulations containing DDM in the early stages of development include intranasal olanzapine plus DDM for the treatment of acute agitation in patients with schizophrenia or bipolar disorder [62].

## 4. Future Perspectives and Conclusions

Intranasal delivery is becoming an increasingly common route of administration for drugs that treat CNS conditions. As of mid-2023, the FDA has approved ten intranasal formulations of CNS drugs as treatments [4,5,6,7,9,10,11,12,13,63], with the three discussed above containing DDM as a permeation enhancer (Figure 8). Intranasal delivery offers several benefits related to ease of delivery, favorable pharmacokinetics, and patient experience. To optimize these benefits, intranasal formulations must address the specific anatomic and physiologic constraints of the nasal cavity, as well as the physicochemical constraints of the drug formulation.

Permeation enhancers such as DDM and TDM are valuable excipients in a growing number of FDA-approved intranasal treatments. They have been shown to enhance drug absorption and bioavailability. DDM has a proven record of preclinical and clinical safety as an excipient in intranasal formulations. Moreover, the physicochemical characteristics of alkylsaccharides offer the potential for the development of other intranasal drug therapies for non-CNS and nonemergency conditions, incorporating hydrophilic and hydrophobic formulations of small molecules, peptides, and proteins. In the CNS, in particular, direct nose-to-brain delivery of drugs with intranasal delivery is an important potential benefit for certain molecules. As real-world experience with these newer intranasal formulations grows, additional potential benefits to patients and healthcare utilization may become apparent. Finally, DDM may also have potential applications in buccal, dermal, oral, and other pharmaceutical agents. While approved intranasal formulations have made progress in providing effective and acceptable treatment for patients with some conditions, significant unmet needs remain for many other conditions that range from allergy to psychiatry. Ongoing research and development of intranasal formulations offers future novel and alternative therapies, thereby broadening the choice of treatment options and improving the experience and outcomes for patients living with a number of acute CNS conditions.

## Figures and Tables

**Figure 1 pharmaceutics-15-02119-f001:**
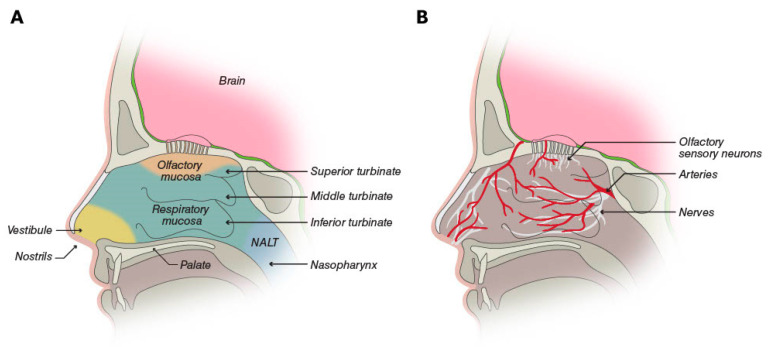
(**A**) The nasal cavity is partitioned into four areas: the vestibule is the area inside the nostrils; the respiratory mucosa and the olfactory mucosa cover the central portion, including the turbinates; and the nasopharynx-associated lymphatic tissue (NALT). (**B**) The respiratory mucosa is the most vascularized part of the nasal cavity, and it is innervated by the trigeminal, palantine, and maxillary nerves. The olfactory mucosa additionally contains olfactory sensory neurons [3]. © Neurelis, Inc. 2023. All rights reserved.

**Figure 2 pharmaceutics-15-02119-f002:**
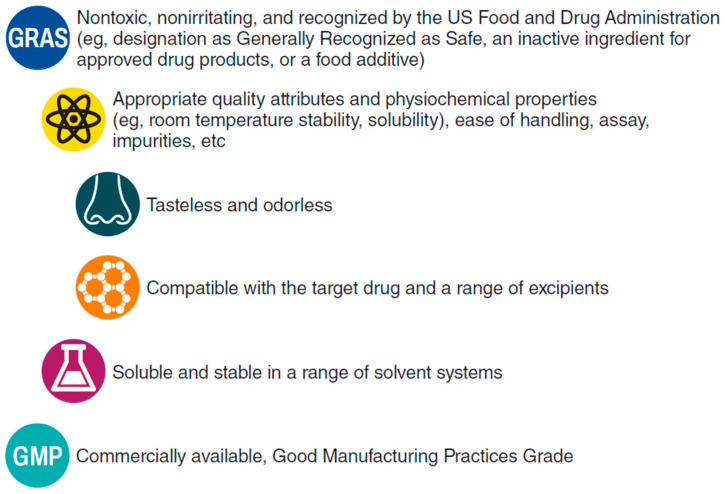
Ideal characteristics of a permeation enhancer for intranasal drug formulations.

**Figure 3 pharmaceutics-15-02119-f003:**
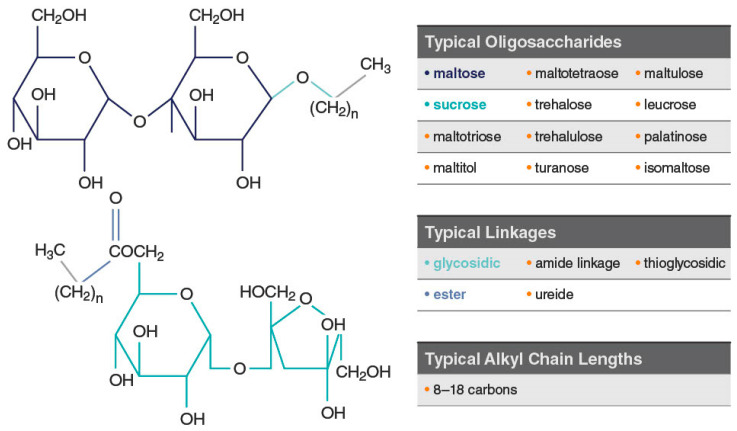
Examples of typical alkylsaccharide structures comprising Intravail^®^ excipients. Used with permission from Maggio & Pillion, 2013 [37].

**Figure 4 pharmaceutics-15-02119-f004:**
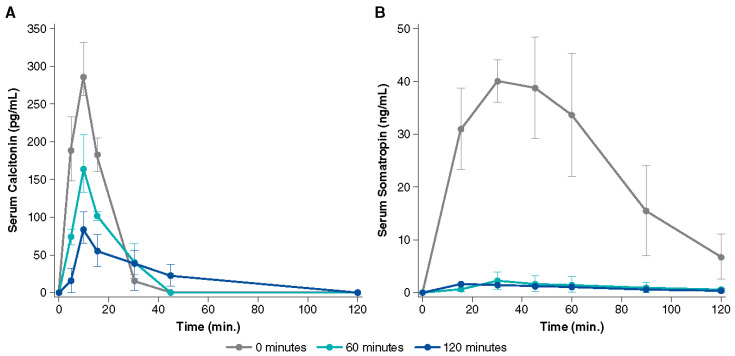
Serum concentration over time after administration of 0.125% tetradecyl maltoside (TDM) with (**A**) calcitonin (2.2 U) and (**B**) somatropin (100 µg) in rats. The uppermost line shows the simultaneous administration of TDM and drug, while the lower lines show administration of TDM alone followed by drug administration after 60 or 120 min. The permeation-enhancing effect of TDM is reversed within 2 h after nasal administration as evidenced by lower serum drug concentration with a longer delay after TDM administration. Used with permission from Arnold et al., 2010 [41].

**Figure 5 pharmaceutics-15-02119-f005:**
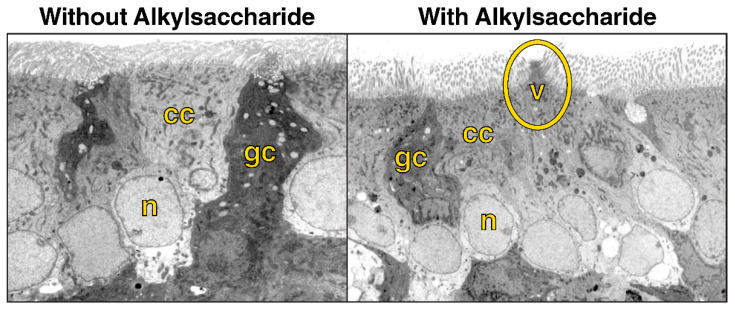
Electron microscopy (original magnification 3500×) of rat nasal mucosa without administration of the alkylsaccharide tetradecyl maltoside (0.125%) and 10 min after alkylsaccharide administration, showing vesicle formation. Note vesicular formation (circled) in right image and cilia at the top of both images. cc, ciliated cell; gc, goblet cell; n, nucleus, v, vesicle. Used with permission from Arnold et al., 2004 [42].

**Figure 6 pharmaceutics-15-02119-f006:**
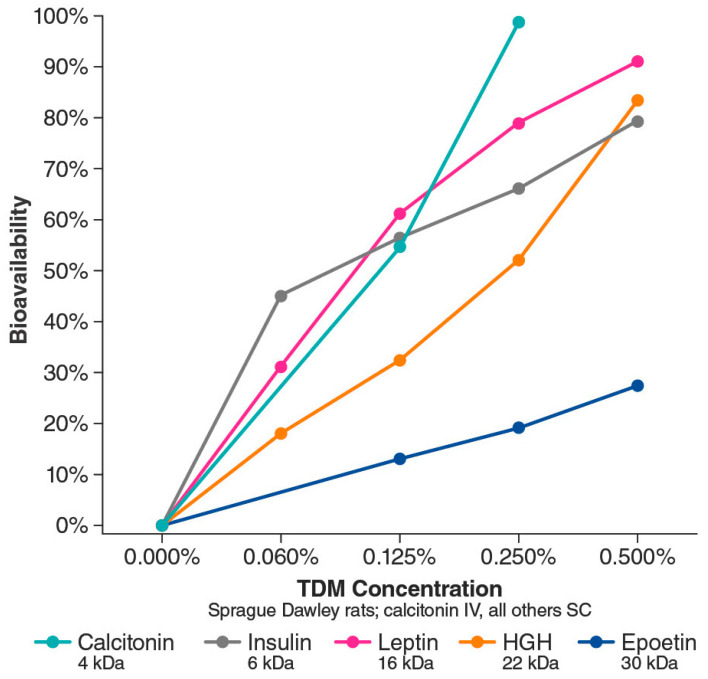
Bioavailability (compared with intravenous delivery) of various intranasally administered proteins ranging from 4 kDa to 30 kDa based on Intravail concentration. HGH, human growth hormone; IV, intravenous; SC, subcutaneous; TDM, tetradecyl maltoside. Adapted with permission from Maggio & Pillion, 2013 [37].

**Figure 7 pharmaceutics-15-02119-f007:**
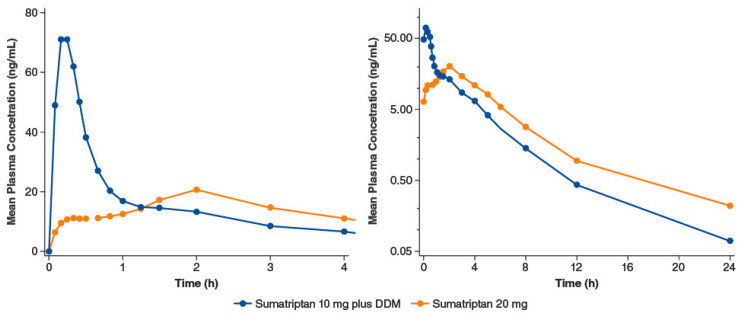
Intranasal sumatriptan 10 mg plus 0.2% dodecyl maltoside (DDM) vs. intranasal sumatriptan 20 mg: mean plasma sumatriptan concentration-time profiles. Panels differ by the span of the x-axis; in addition, the y-axis of the right panel is on a log scale. Adapted with permission from Munjal et al., 2016 [47].

**Figure 8 pharmaceutics-15-02119-f008:**
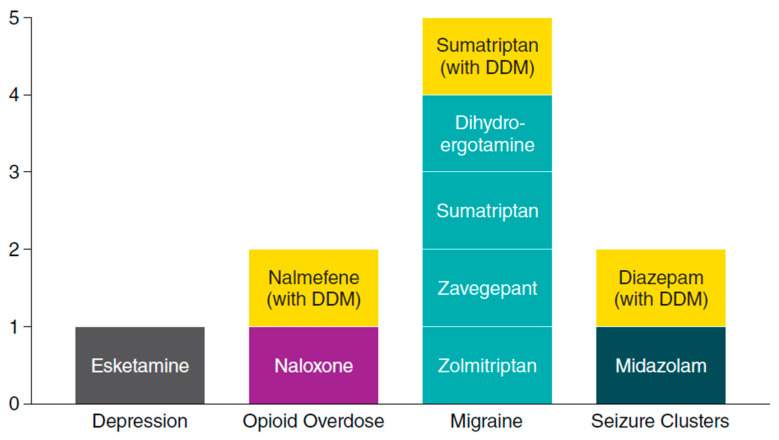
Central nervous system drugs approved by the US FDA as intranasal formulations by mid-2023. CNS, central nervous system; DDM, dodecyl maltoside [4,5,6,7,9,10,11,12,13,63].

## Data Availability

Not applicable.

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
