# Peer review of "Optimizing Absorption for Intranasal Delivery of Drugs Targeting the Central Nervous System Using Alkylsaccharide Permeation Enhancers"

_pharmaceutics, 2023, doi:10.3390/pharmaceutics15082119_

Round 1
Reviewer 1 Report
The manuscript entitled “Optimizing Absorption for Intranasal Delivery of Drugs Targeting the Central Nervous System” aims to review factors related to the delivery of intranasal drugs and the role of alkylsaccharide permeation enhancers in the context of approved and future intranasal formulations of drugs for CNS conditions. The topic is very interesting, and the manuscript is well-prepared in general. Also, the manuscript is within the scope of the journal. Still, some minor issues need to be addressed in order for the paper to be published. My specific comments are given below.
The abstract is suitable.
The introduction is very short and not informative enough. It needs to address the problem in more detail, explain the authors’ motivation for writing this paper and claim the novelty of the manuscript.
The main body of the manuscript is well-prepared.
It would be highly beneficial to include a separate section on future perspectives regarding this topic, with a critical view of the advantages and shortcomings of this type of CNS disease therapy.
The conclusion needs to be improved. It is too short and should provide more details regarding this topic's current state and future expectations. The importance of addressing this issue should also be stated.
The literature is up-to-date.
The language is suitable.
Reviewer 2 Report
This is amazing manuscript on an interesting topic such as intranasal drug delivery. I believe the table and the figures are appropriate and literature is up to date. Also this study will add to the body of literature in drug delivery field.
Reviewer 3 Report
The authors of the manuscript have compiled a review of the drug delivery from the nose to the brain, focusing on alkylsaccharides. Three formulations that are commercially available in the United States of America and contain dodecyl maltoside (DDM) are presented in detail. The manuscript's structure is logical and the quality of the figures is adequate.
Critical comment:
In section 2.2.1, the authors write that droplet size ideally means particles of 10 um or more in diameter. Is there any information on the upper size limit?
A graph would be useful. How many intranasal formulations approved by the FDA are of the nose-to-brain type.? Also, of those formulations, what % currently use DDM in the formulation?
In addition to the 3 formulations presented above, are there any known intranasal formulations approved for the European Union that contain DDM?
Reviewer 4 Report
The manuscript is interesting and worth considering for publication. However, Authors should pay attention to the following aspects:
1) The font of the table content is not in line with the requirements of the Journal. Additionally, it will be clearer to present these data in the form of graphics.
2) The paper contains some abbreviations hence it is suggested to supplement the manuscript with additional subsection containing all abbreviations and their explanations.
3) Section References should be significantly extended (including mainly more up-to-date references) - 59 references are definitely not enough when it comes to the review paper. Additionally, this section should be prepared in line with the requirements of the Journal, e.g., reference [32] contains the whole journal name instead of its abbreviation.
The language of the paper is appropriate.
Reviewer 5 Report
The manuscript entitled “Optimizing Absorption for Intranasal Delivery of Drugs Targeting the Central Nervous System” deals the detailed review about the positive effect of dodecyl maltoside and tetradecyl maltoside on the nose-to-brain transport of drugs. The manuscript is well-written, collects the novel information about two commonly applied absorption enhancer, however based on the title the reader would love to here about other advantageous approaches as well, as nano carriers, in situ gelling polymers, or medical devices, which are able to control the deponation to olfactory region. Please also provide a short and concise paragraph of each to cover most of the field possible approaches, which can be exploited for targeting the CNS.
